# Support Needs of Children with Autism Spectrum Disorders: Implications for Their Assessment

**DOI:** 10.3390/bs13100793

**Published:** 2023-09-24

**Authors:** Verónica M. Guillén, Miguel Á. Verdugo, Pedro Jiménez, Virginia Aguayo, Antonio M. Amor

**Affiliations:** 1Department of Education, Universidad de Cantabria, 39005 Santander, Spain; 2Institute for Community Inclusion and Department of Personality, Assessment, and Psychological Treatments, Universidad de Salamanca, 37005 Salamanca, Spain; verdugo@usal.es (M.Á.V.); aguayo@usal.es (V.A.); aamor@usal.es (A.M.A.); 3Institute for Community Inclusion, Universidad de Salamanca, 37005 Salamanca, Spain; pedrojimenezn@gmail.com

**Keywords:** autism spectrum disorder (ASD), intellectual disability (ID), supports intensity scale for children (SIS-C), support needs, assessment

## Abstract

The construct of support needs has become a key aspect for the diagnostics, classification, and interventional management of autism spectrum disorders (ASDs). However, instruments specifically designed to assess support needs in this population are not available. Currently, the Supports Intensity Scale for Children (SIS-C), which could be administered to assess students with any type of intellectual disability (ID), is the only valid tool able to assess support needs in Spain. Our aim was to verify whether the SIS-C is useful for assessing the support needs of students with ASD, regardless of whether or not they present ID. The participants were subdivided into two groups. One group included students with ASD and ID (*n* = 248), and the other comprised participants with ASD without an ID (*n* = 44). The results of the two groups were compared with those reported in the original validation sample of the SIS-C, which involved participants with ID without ASD (*n* = 566). The results showed that this scale could be useful for assessing support needs in the three subgroups, but it appeared that different standardized norms based on the characteristics of each specific population would be appropriate.

## 1. Introduction

The incidence of the continuum of conditions that represents autism spectrum disorders (ASDs) in the global population is currently high. Recent studies indicated that the ASD incidence ranges between 1 in 36 [1] and 1 in 100 births [2]. Although the ASD prevalence reported in these studies is highly variable and could diverge even more considering the geographical context [3], most studies showed a global increase in the prevalence rate of ASD. This is particularly relevant for the subgroup of patients without intellectual disability, which includes most people with an ASD diagnosis, classified as having an average intelligence. Currently, although ID is the disorder most commonly co-occurring with ASD, it is present in only 33–38% of cases [1,2]. Moreover, several studies showed that less than 5% of ASD cases have an above-average intelligence [4,5].

Despite ID being a strong predictor of poor prognosis [6], there is great consensus about the fact that all individuals with ASD, even those with better intellectual functioning, exhibit deficits in their adaptive skills and functioning abilities [7]. Specifically, some studies indicated that ASD mainly affects ‘executive functions’ [8], whose main components (initiation, organization, and working memory) are closely related to the development of adaptive behavior [9]. The variability in cognitive level and daily living skills is also associated with a wide heterogeneity in the support needs of this population. There is large evidence of their support needs in fields related to physical, mental, and behavioral health [10,11] in both home and community contexts [12]. However, studies on the assessment of support needs—an important construct of the new ASD definition according to the Diagnosis Statistical Manual (DSM) [13,14]—are very recent and mainly focused on people with ASD and ID [15,16,17].

The difficulties in the inclusion of people with ASD in the community are very diverse and mainly regard school inclusion [18]. According to teachers [19], students with ASD usually exhibit support needs closely related to participation and learning, even though their IQ is within a normal range, which could be due to additional deficits besides those in the executive function, such as in central coherence, mirror neuron system development, and the theory of mind [20,21,22,23]. Memory and metamemory problems are often also present [24], which could affect the development of their academic skills [25,26,27].

Despite evidence that ASD students have school support needs, most of the ASD management strategies regard ASD diagnosis and classification rather than support planning [28,29]; consequently, there are still high rates of both school discrimination [30] and academic failure for students with this condition (with and without an ID), in both secondary and baccalaureate schools [26]. Although some attempts have been made to establish the psychoeducational profiles of students with ASD with the aim to develop specific educational plans [31], it is necessary to establish more complete and integrated support systems, adjusted to all the characteristics and needs of students with ASD, relying on evidence-based strategies and practices [32,33], and going beyond specific behavioral goals to also focus on ASD students’ participation in classrooms, institutions, and the community [34]. Recent studies [35,36] pointed out that there is poor provision of individualized support in schools, especially at fundamental moments of students’ development such as during transitions between educational stages and in preparation for adult life. This is due to the lack of tools for assessing the support needs of individuals with ASD in all contexts, which would allow for comprehensive and personalized support planning [28,29].

Currently, despite the evident importance of defining the support needs of children and adolescents with ASD for their diagnosis and classification as well as for designing effective personalized interventions, we do not have any specific instrument for their assessment. However, thanks to the change in the conception of disability that began in the last century and emphasizes a socio-ecological and multidimensional approach [28], organizations that work with people with disabilities have introduced individualized plans and are now developing their assessments and interventions according to the new Quality of Life Supports Paradigm (QOLSP) [37]. In this perspective, support is conceived as a bridge between personal skills and environmental demands [29], defined by the activities and contexts of an inclusive society in which all individuals, with and without disabilities, are present, interact, develop their skills, and can improve their quality of life.

This approach emphasizes the individuals’ relationship with their environment, overcoming traditional visions focused on their deficits and highlighting the need to reduce the inequalities they experience in their social contexts [28]. Consistently with this conceptual approach, different standardized measures have been developed to evaluate the discrimination experienced by people with disabilities in order to elaborate personalized support plans. The first tool designed for this purpose is the Supports Intensity Scale (SIS) [38,39], created by the American Association on Intellectual and Developmental Disabilities (AAIDD) and validated internationally in thirteen countries [40], specifically intended for the assessment of the support needs of adults with IDs. The benefits obtained from the use of this instrument [41] focus on enhancing the distribution of organizational resources and the development of personalized support plans to improve the quality of life of the recipients [42], which almost immediately led organizations to demand a similar tool adapted for children. Such an instrument should consider the school environment in order to optimize the participation of children with intellectual and developmental disabilities and their interactions with their peers.

Thus, the Supports Intensity Scale for Children (SIS-C) was created by the AAIDD [43,44] and immediately adapted to the Spanish context [45] through a rigorous process of translation and adaptation, following the steps proposed by Tassé and Craig [46] and the requirements of the International Test Commission (ITC) [47]. These guidelines are aimed at ensuring not only linguistic translation but also a semantic, conceptual, and cultural adaptation.

The SIS-C gathers information on support needs in different daily life contexts and is scaled for different age groups between 5 and 16 years [48], allowing the determination of the specific continuum level of the individuals assessed in comparison with their peers (i.e., other children of the same age who have IDs). Numerous studies tested the SIS-C psychometric suitability both internationally [49] and in the Spanish context [45,50]. After the multiple applications of this tool to support students with intellectual and developmental disabilities, its specific operation in the child and adolescent population with ASD is currently an incipient research challenge.

This study aimed to test the following hypotheses: (1) the SIS-C may be as reliable and valid when exclusively applied within the population of individuals with ASD, in comparison to the population of individuals with general intellectual and developmental disabilities; (2) the SIS-C might be relevant and provide potential benefits when applied to children with ASD regardless of an associated ID diagnosis; and (3) there may be significant variations in the required support intensity among children with IDs, depending on whether they have ASD.

## 2. Materials and Methods

### 2.1. Participants

As can be seen in Table 1, the present study included a group of children and adolescents aged between 5 and 16 years, with an ID and without ASD (*n* = 566), and two groups of participants with ASD, one including 248 individuals with ASD and an ID, and the other comprising 44 individuals with ASD without an additional diagnosis of ID.

It can be observed that in the three participating subgroups, the percentage of males was higher than that of females, which is representative of the statistical incidence of the disorder according to sex [2]. With respect to the mean age, it was slightly higher in the ID group without ASD. The percentage of children attending special education schools was also higher in the ID group without ASD (66.19%), which is related to the high percentage of individuals with severe and profound disabilities within that group (33.45%). No individuals in the third group attended special education schools. In addition to the data shown in the table, we highlight that most of the participants lived with their families (95%) or in reception centers (3%). Spanish was the predominant mother tongue in the three sample groups (95%), followed by Galician (2%).

Within the group of participants with an ID and without ASD, there was a predominance of individuals with unknown etiologies (*n* = 317), Down syndrome (*n* = 111), and cerebral palsy (*n* = 101). In addition, in the two groups of individuals with ASD, the majority of the cases had no other associated diagnoses, although some of them were also diagnosed with attention deficit hyperactivity disorder (ADHD). In any case, it is relevant to highlight that in the group of children with ASD and an ID, the number of individuals with speech and language alterations was very high (32%), with approximately 10% of them needing physical instruments, electronic supports, and/or some type of alternative and/or augmentative communication system.

### 2.2. Instrument

The SIS-C is the only support needs assessment tool that allows for the reliable and valid assessment of the intensity of extraordinary support (i.e., linked to the disability and not to evolutionary development) needed by children and adolescents with IDs in their vital contexts. This study used the validated Spanish version of the scale [45,50].

The SIS-C is composed of two main sections. The first section includes descriptive items related to exceptional medical and behavioral needs, whereas the second, which composes the fundamental part of the scale from which to draw an index and profile of the support needs, consists of 61 items related to seven areas of everyday life: (a) home life (9 items); (b) community and neighborhood (8 items); (c) school participation (9 items); (d) school learning (9 items); (e) health and safety (8 items); (f) social activities (9 items); and (g) advocacy/self-representation (9 items). Each item must be scored according to three measurement indices (type of support, frequency of support, and daily time of support), using a five-point (0–4) Likert-type rating scale, where greater scores always represent a higher intensity of the needed supports. This instrument should be employed by individuals who know the children being evaluated well (e.g., professionals and family members), either through semi-structured interviews conducted by experts or autonomously, after receiving specific training on the scale.

### 2.3. Procedure

The sample of the present study corresponded, in large part, to the general sample of the study of adaptation and validation of the SIS-C in the Spanish context [45]. The participants were 814 children and adolescents with IDs attending more than fifty schools, organizations, and/or entities that provided services to individuals with IDs throughout Spain, who were contacted via email. The confidentiality of the data obtained was ensured at all times, and the criteria set by the Bioethics Committee of the University of Salamanca for social research were met. The resolution is available upon request to the first author.

The requirements that the children had to meet to be assessed with the SIS-C were the following: (a) being between 5 and 16 years old; (b) exhibiting an ID (of any degree); and (c) having professional and/or family members who had known the children for more than three months and who, on a voluntary basis, would answer the SIS-C questions regarding the children’s support needs.

The sample selection method was incidental sampling, due to the obligation to have the informed consent signed from both the interviewees and the legal representatives of the students assessed. The criteria suggested by the AAIDD for the adaptation of a scale to other contexts were followed, distributing the total sample in a balanced way and taking into account the variable of age (groups of 5–8, 9–12, 13–16 years) and the level of ID (mild, moderate, severe/profound). The etiology of IDs was very diverse, and their classification specifically defined a subsample of 248 participants with ASD. Although these students were part of the total sample used to validate the SIS-C and it was proved that this scale is reliable for assessing the support needs of any children or adolescents with an ID, the specific characteristics of the individuals with ASD made the analysis of the scale behavior necessary when it was applied to individuals of this population, with or without an ID.

Specifically, our sample of individuals with ID was the general validation sample of the SIS-C. The data were thus obtained from the 248 participants included in the validation of the scale who had an ID and ASD (the diagnosis of ASD was verified through clinical records) and from 566 children and adolescents who were also part of the general validation, but whose ID did not have a specific etiology or was associated with other disorders different from ASD. In addition, the SIS-C was administered to 44 children with ASD without an ID in order to analyze if it could also be used to assess the support needs in this population.

### 2.4. Data Analyses

Considering the central hypotheses of the present study, it was first necessary to determine the functioning of the scale when applied to the participants with ASD (with or without an ID). To that end, the reliability (Cronbach’s Alpha and inter-rater reliability) and validity (construct and criteria) of the scale with respect to the data from the participants with ASD were determined.

Subsequently, we analyzed whether there were significant differences in the scores obtained in the SIS-C by students with an ID and ASD and students with an ID without ASD. Then, we assessed the performance of the scale when applied to students with ASD and without an ID. The descriptive data obtained from the three subgroups of the sample are presented, as well as the results of the relevant means of contrast tests performed with the IBM SPSS Statistics software (v.26). After checking the data for normality, non-parametric tests were implemented. Spearman’s rho was the coefficient selected to measure correlations, and the Mann–Whitney *U* test was used for the mean contrast tests.

## 3. Results

### 3.1. Reliability and Validity of the Scale in a Population with Autism Spectrum Disorder

Regarding the scale’s reliability, two criteria were taken as a reference: Cronbach’s alpha (n = 248) and inter-rater reliability (based on the 246 cases for which a double assessment was conducted). They were used to assess the performance of both the entire scale and each of its sections. As can be seen in Table 2, for both criteria, results above 0.700 were obtained in all areas, indicating good reliability; internal consistency could be considered optimal (Cronbach’s alpha greater than 0.900).

With respect to the scale’s validity, as previously performed for the general validation of the SIS-C, construct and criterion validity were assessed. For construct validity, three different correlations were used: (1) the score of each subscale and its corresponding items, (2) the scores of each subscale, and (3) the score of each subscale and the total score of the scale. In order to assess the validity of the criteria, all the professionals who completed the SIS-C for its validation were asked to provide an external opinion (based on observations, experiences, reports, etc.) regarding the support needs of the participants before applying the instrument, which was then compared with the results of the SIS-C. As can be seen in Table 3, all the correlations obtained in the construct and criterion validity assessments were greater than 0.600, indicating an acceptable validity.

### 3.2. Differences in the Results of the SIS-C When Comparing Children with (1) Intellectual Disability and Autism Spectrum Disorder, (2) Intellectual Disability without Autism Spectrum Disorder, and (3) Autism Spectrum Disorder without Intellectual Disability

To address the second goal of this study, the descriptive data obtained from the three examined subsamples are reported.

As can be seen in Table 4, in all cases, the scores were higher for the participants with an ID and ASD, the difference being especially relevant in the areas of (1) social activities, (2) school participation, and (3) advocacy (self-representation). On the other hand, in the areas of school learning and home life, the differences between the subgroups were much smaller.

In order to know whether the observed differences were significant, the Mann–Whitney *U* mean rank contrasts were evaluated. First, we compared data from students with an ID and ASD and students with an ID not associated with ASD. The differences were significant (*p* < 0.05) between the groups for both the general scores (*z* = 5.447; *p* < 0.001) and the scores obtained for (a) home life: *z* = −2.154; *p* = 0.031; (b) community and neighborhood: *z* = 4.240; *p* < 0.001; (c) school participation: *z* = 5.495; *p* < 0.001; (d) school learning: *z* = 3.439; *p* < 0.001; (e) health and safety: *z* = 5.130; *p* < 0.001; (f) social: *z* = 8.719; *p* < 0.001; and (g) advocacy: *z* = 5.575; *p* > 0.001.

After establishing the differences between the subgroups with an ID (with and without ASD), the functioning of the scale was analyzed through applying it to the data from the 44 students who exhibited ASD without an ID. As it is also shown in Table 4, the children with ASD and high functioning were classified as exhibiting ‘extraordinary support needs’ in all areas of daily life considered in the SIS-C, that is, not only in those areas more related to social life, although their scores were much lower than those of children with ASD and an ID.

It is also important to highlight that, the children with an ID (with or without ASD) exhibited greater support needs in the area of school learning, whereas the children with ASD but without an ID appeared to require special support in the social area. Another Mann–Whitney *U* mean rank contrast test was performed to compare students with ASD and an ID and students with ASD but without an ID. We found that the scores of the two groups were significantly different for (a) home life: *z* = −7.167; *p* < 0.001; (b) community and neighborhood: *z* = −7.314; *p* < 0.001; (c) school participation: *z* = −7.910; *p* < 0.001; (d) school learning: *z* = −8.372; *p* < 0.001; (e) health and safety: *z* = −7.803; *p* < 0.001; (f) social: *z* = −6.653; *p* < 0.001; and (g) advocacy: *z* = −7.042; *p* < 0.001.

As can be observed, the differences were greater than those found between the group with ASD and an ID and the general group with IDs for all areas, except for the area of social activities. In fact, the means obtained for the two groups with ASD in the area of social activities were similar, and, therefore, their difference was slightly low, though it remained significant.

## 4. Discussion

Improving the quality of life of people with disabilities is an important objective of social policies. To this aim, it is necessary to make a correct assessment in order to offer personalized supports, especially to young people and children.

In spite of the importance and necessity of having appropriate tools for correctly assessing the needs of people with disabilities to improve their quality of life and social inclusion, little research has hitherto been conducted to develop support needs scales for assessing children with intellectual disabilities. In this sense, the recent adaptation and validation of the SIS-C [45] is a significant contribution to this field. From its publication, several studies have revealed multiple trajectories of research and applications of knowledge in the assessment of support needs for individuals with IDs [51].

Additionally, studies should point out the importance of identifying the support needs of specific populations and facilitate the individualized design and implementation of educational and social inclusion programs. As regards the assessment of children with ASD, there is still a wide need for appropriate tools focused on improving support planning, as most research in this field is focused on disorder diagnosis and classification [52].

It is known that students with ASD can learn constantly, although sometimes they do not know how, cannot, or do not have the opportunity to demonstrate their progress. This fact is mainly due to the lack of adequate means to assess their skills in relation to the environmental demands or to the rigidity of the systems/individuals that assess this progress. Despite the evidence of the importance of assessing the support needs of children with ASD for school-targeted interventions [18,53] and progress evaluation, no instruments with proven effectiveness are available. Therefore, the aim of the present study was to determine whether all children and adolescents with ASD (with or without an ID) could benefit from the application of the SIS-C to evaluate their specific needs.

During the process of the creation and validation of the SIS-C, several studies were examined in relation to the psychometric properties and utility of the scale. We found increasing evidence of the good psychometric performance of the SIS-C [49], as shown by studies conducted to obtain distinct standardized norms differentiated by age group [48]. Likewise, as already observed for the SIS version for adults [54,55,56,57], it appeared necessary to assess the functioning of the SIS-C in specific groups, for example, in children with cerebral palsy [58] or ASD with an ID [15,16]. The assessment of the functioning of the SIS-C in a population with ASD indicated good reliability and validity, which allowed us to conclude that the use of the SIS-C was adequate for this population. It can be seen that the results obtained in the present study are similar to those obtained for the general validation of the instrument in Spain [45,50].

As expected, considering previous studies addressing the limitations of individuals with ASD (with or without an ID) (e.g., [12,59]), the SIS-C included items and contexts in which all students with an ID and/or ASD exhibited support needs in a relevant way. However, it should be noted that the individuals with an ID and ASD scored significantly higher in support needs in all sections than individuals with IDs with other etiologies and individuals with ASD without an ID. These scores were also distributed differently with respect to the vital contexts for the three proposed subgroups, thus indicating the need to use differentiated scales for these specific populations to achieve an optimal functioning of the instrument. Thus, for individuals with IDs, the school learning context always entailed very high scores. On the other hand, individuals with ASD, especially those without an ID (although they still needed extraordinary support in this context) exhibited greater needs in other social and self-determination areas, a finding that is in line with the conclusions reported in other studies [60].

The findings of the present study should be considered preliminary, and some limitations must be identified. Firstly, although we collected sufficient evidence supporting the use of the SIS-C in the assessment of children with ASD (with or without an ID), we need to be cautious in the interpretation of the resulting data. It is important to point out that, although the quality indices were adequate when taking into account only ASD children, they were slightly lower when considering children with IDs of other etiologies [45], a fact that could have been influenced by the smaller sample size of the present study in comparison to the original and/or to the wide heterogeneity of ASD conditions. Similarly, introducing some item modifications to better represent the specific characteristics of this condition could improve the functioning of the scale. Specific norms should also be created for this population in order to acquire more adjusted information. Lastly, we cannot forget when interpreting the results that they were derived from observed scores which, as such, were not free from bias.

The purpose of any support needs assessment should be to ascertain as accurately as possible the extraordinary support needs of individuals with respect to their reference groups in order to be able to design and implement individualized support plans [29], provide the necessary resources efficiently, and intervene in a comprehensive and effective manner [61]. For this reason, it is necessary to continue conducting studies in this line, adding more participants and performing multigroup confirmatory factor analyses to determine whether there is a measurement invariance that ensures a study of unbiased comparisons of latent means. In this way, it will be possible to better assess the functioning of the scale, carry out rigorous and specific comparisons (item by item) between subgroups, and create specific scales capable of collecting and reflecting the characteristics of young individuals with ASD, the ultimate goal being to improve their quality of life and inclusion in the community.

## Figures and Tables

**Table 1 behavsci-13-00793-t001:** Sociodemographic data of the three study groups.

		ID (no ASD) (*n* = 566)	ASD and ID (*n* = 248)	ASD (no ID) (*n* = 44)
Gender	Male	335	193	40
	Female	231	55	4
Age	5–8 years old	136	82	15
	9–12 years old	157	91	15
	13–16 years old	273	75	14
ID Level	NO ID	0	0	44
	Mild	162	44	0
	Moderate	210	80	0
	Severe	134	61	0
	Profound	56	9	0
	Missing data	4	54	0
Schooling	Regular school	142	37	41
	Special school	375	118	0
	Combined education	40	99	2
	Not reported	9	4	1

Note: ID = intellectual disability; ASD = autism spectrum disorders.

**Table 2 behavsci-13-00793-t002:** Reliability of the SIS-C; sample with ASD.

Subscales	Cronbach’s Alpha	Inter-Rater Reliability
A. Home Life	0.935	0.901 **
B. Community and Neighborhood	0.947	0.764 **
C. School Participation	0.927	0.773 **
D. School Learning	0.948	0.727 **
E. Health and Safety	0.929	0.781 **
F. Social	0.933	0.772 **
G. Advocacy	0.945	0.773 **
Total	0.985	0.846 **

** Correlation is significant at the 0.01 level.

**Table 3 behavsci-13-00793-t003:** Validity of the SIS-C; sample with ASD.

Subscales	Construct Validity			Criteria Validity
	Range of Item–Subscale Correlations	Range of Subscales Correlations	Range of Subscales–Total Correlations	
A0. Home Life	0.787–0.862 **	0.600–0.793 **	0.860 **	0.831 **
B0. Community and Neighborhood	0.830–0.869 **	0.733–0.793 **	0.888 **	0.671 **
C0. School Participation	0.649–0.912 **	0.747–0.822 **	0.908 **	0.700 **
D0. School Learning	0.778–0.882 **	0.600–0.758 **	0.839 **	0.642 **
E0. Health and Safety	0.795–0.880 **	0.704–0.864 **	0.911 **	0.712 **
F0. Social	0.712–0.884 **	0.718–0.861 **	0.915 **	0.646 **
G0. Advocacy	0.819–0.909 **	0.695–864 **	0.903 **	0.659 **

** Correlation is significant at the 0.01 level.

**Table 4 behavsci-13-00793-t004:** Descriptive statistics for the SIS-C scores of the three examined groups: children with an ID and ASD (*n* = 248), an ID without ASD (*n* = 566), and ASD without an ID (*n* = 44).

Subscales	Group	Mean	Standard Deviation	Median	Mean Rank
A. Home Life	ID with ASD	57.73	23.74	61.00	54.00
	ID without ASD	52.91	32.36	49.00	54.00
	ASD without ID	25.00	21.28	17.00	46.50
B. Community and Neighborhood	ID with ASD	67.28	16.67	71.00	50.50
	ID without ASD	57.06	26.86	61.50	48.00
	ASD without ID	34.39	26.04	31.00	48.00
C. School Participation	ID with ASD	75.70	20.27	78.00	57.00
	ID without ASD	62.32	30.69	64.00	54.00
	ASD without ID	34.34	27.97	31.50	46.00
D. School Learning	ID with ASD	87.51	17.14	90.00	59.50
	ID without ASD	79.52	24.55	83.00	54.00
	ASD without ID	43.18	29.34	38.00	50.00
E. Health and Safety	ID with ASD	69.60	18.35	72.00	53.00
	ID without ASD	59.26	25.67	61.00	48.00
	ASD without ID	30.55	22.83	23.00	38.50
F. Social	ID with ASD	81.09	19.78	84.00	58.50
	ID without ASD	60.98	30.45	61.00	54.00
	ASD without ID	48.57	27.55	42.50	48.50
G. Advocacy	ID with ASD	80.66	19.92	81.00	62.00
	ID without ASD	67.55	29.13	69.50	54.00
	ASD without ID	44.91	27.33	40.50	47.50
Total	ID with ASD	515.79	126.38	534.00	422.50
	ID without ASD	439.62	186.84	445.50	379.50
	ASD without ID	262.24	172.97	205.50	316.00

## Data Availability

Data are available upon request to the corresponding author.

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
