# Peer review of "Support Needs of Children with Autism Spectrum Disorders: Implications for Their Assessment"

_behavsci, 2023, doi:10.3390/bs13100793_

Round 1

Reviewer 1 Report

Dear Authors,

Thank you for the possibility of revising this interesting article on support needs in a sample of autistic children with and without intellectual disability. 

This work covers an important topic and may be important especially in the clinical context of ASD. 

However, several significant issues need to be addressed before the official publication in Behavioral Sciences Journal. 

Introduction

  • I would deepen the part in which statistical sustainability was discussed in both the American and Spanish contexts. This is a very crucial and relevant part that is worth mentioning.

  • The authors mention several research questions, however, precise and coherent hypotheses should follow the broader research questions. I would suggest the authors rephrase the final part of the introduction with more structured hypotheses.

Materials and Methods

  • More information about the sampling procedure should be indicated in the procedure part

  • Did the participants sign an informed consent? 

  • How is ASD assessed in the participants in this group?

  • Additional general information is needed in this section to avoid confusion

Data analysis

  • An analytic plan should be reported at the beginning of this section indicating the steps of the analysis

  • Did the authors check data for normality before implementing the parametric tests?

Discussion

  • This section should be deepened and discussed in light of the findings and the potential limitations of the current study 

Massive English proofreading is needed throughout the manuscript.

Author Response

Introduction

  • I would deepen the part in which statistical sustainability was discussed in both the American and Spanish contexts. This is a very crucial and relevant part that is worth.

Answer: a sentence has been added:

Thus, the Supports Intensity Scale for Children (SIS-C) was created by the AAIDD [43,44] and immediately adapted to the Spanish context [45] through a rigorous process of translation and adaptation, following to the steps proposed by Tassé and Craig [46] and the requirements of the International Test Commission (ITC) [47]. These guidelines are aimed at ensuring not only a linguistic translation, but also a semantic, conceptual, and cultural adaptation.

  • The authors mention several research questions, however, precise and coherent hypotheses should follow the broader research questions. I would suggest the authors rephrase the final part of the introduction with more structured hypotheses.

Answer: Research questions have been rewritten as hypotheses.

 Materials and Methods

  • More information about the sampling procedure should be indicated in the procedure part

Answer: A sentence has been added regarding this information.

  • Did the participants sign an informed consent? 

Yes. It was been better explained.

  • How is ASD assessed in the participants in this group?

Answer: Diagnosis of ASD was verified through clinical records. This sentence has been included in the manuscript.

  • Additional general information is needed in this section to avoid confusion

Answer: This explanation has been added:

Specifically, our sample of individuals with ID was the general validation sample of the SIS-C. The data were thus obtained from the 248 participants included in the validation of the scale who had ID and ASD (the diagnosis of ASD was verified through clinical records) and from 566 children and adolescents who were also part of the general validation, but whose ID did not have a specific etiology or was associated with other disorders different from ASD. In addition, the SIS-C was administered to 44 children with ASD without ID in order to analyze if it could also be used to assess the support needs in this population.

Data analysis

  • An analytic plan should be reported at the beginning of this section indicating the steps of the analysis

Answer: Done

  • Did the authors check data for normality before implementing the parametric tests?

Normality was checked for total scores. After checking normality for each scale and found a big lack of normality in some areas, we decided better used non-parametric tests. This part has been explained in the text.

Answer: After checking data for normality, non-parametric tests were finally implementing. Spearman Rho was the coefficient selected for correlations and the Mann-Whitney U was used for the mean contrast tests.

Discussion

  • This section should be deepened and discussed in light of the findings and the potential limitations of the current study 

Answer: Sentences in discussion have been modified in order to follow this suggestion.

  • We also sent the manuscript to the proof-reading service recommended for the editors of this journal -

Reviewer 2 Report

The article can be published, topic Support Needs of Children with Autism Spectrum Disorders: Implications for Assessment, is of scientific and topical interest; is close to the special issue of the Behavioral Sciences Journal; I congratulate you on this choice and the research you have done. Regarding the elaboration of the manuscript, in my opinion, there are several proposals for improvement:

- The number of subjects that were included in the research should be clarified; according to the data in Table no. 1. Sociodemographic data of the three study groups, there are 858 subjects (..."with ID and without ASD (n = 566), and two groups of participants with ASD: (1) 248 individuals with ASD and ID; and (2) 44 individuals with ASD and without additional diagnosis of ID”) (lines 116, 117 and 118); in the research, 814 subjects are analyzed (see line 159), and later, in data processing, the number 813 appears (line 229). In my opinion, more details should be specified about how the subject groups were included in the various researches and comparisons, which are comprehensive and with interesting results.

I would recommend that more arguments regarding the obtained results be given in the article, considering the fact that more categories of subjects were researched. A longer bibliography could help in this direction. Also, the results should be analyzed and the conclusions should be compared more from the perspective of the results of other studies, especially because they used instruments applied to subjects from other countries and different contexts.

            Improving the quality of life of people with disabilities is an important objective in social policies systems; that is why it is very important to make a correct assessment in order to offer support to this population category, especially young people and children. The results of the study should better point out the need for these studies which would also be useful from the perspective of educational integration and social inclusion, especially for children and young people; I recommend looking at other articles as well, such as: https://doi.org/10.3390/su13137056

https://doi.org/10.3390/su13052558

https://doi.org/10.3390/su14084630.

            The presentation of a section with reference to the limits of the study, would demonstrate honesty and frankness in the investigation, but also implications for future research.

            The conclusions of the study should show the importance and necessity of correctly assessing the needs of people with disabilities, with appropriate tools, to improve their quality of life, integration and social inclusion.

Author Response

The article can be published, topic Support Needs of Children with Autism Spectrum Disorders: Implications for Assessment, is of scientific and topical interest; is close to the special issue of the Behavioral Sciences Journal; I congratulate you on this choice and the research you have done. Regarding the elaboration of the manuscript, in my opinion, there are several proposals for improvement:

- The number of subjects that were included in the research should be clarified; according to the data in Table no. 1. Sociodemographic data of the three study groups, there are 858 subjects (..."with ID and without ASD (n = 566), and two groups of participants with ASD: (1) 248 individuals with ASD and ID; and (2) 44 individuals with ASD and without additional diagnosis of ID”) (lines 116, 117 and 118); in the research, 814 subjects are analyzed (see line 159), and later, in data processing, the number 813 appears (line 229). In my opinion, more details should be specified about how the subject groups were included in the various researches and comparisons, which are comprehensive and with interesting results.

Answer: 814 was the number of people participating in the Spanish validation of the scale. For this validation is was necessary that all children had ID. From this sample, we got 248 children with ID and ASD (the other 566 had a different diagnosis). After that, for this study, we also administered the scale to 44 children with ASD but without ID. That’s is why the total sample is 866 but in line 159 we talk about 814 (in this part we are just talking about the validation).  In any case, we have added a sentence (in red) for make it clearer:

Specifically, our samples of individuals with ID formed the general validation sample of the SIS-C. Data was thus obtained from the 248 participants included in the validation of the scale who had ID and ASD, and from 566 children and adolescents who were also part of the general validation, but whose ID did not have a specific aetiology or was associated with other disorders other than ASD. Complementary, SIS-C was administered to 44 children with ASD but without ID in order to analyse if the scale could also be used to assess support needs in this population.

The number 813 was related to the degrees of freedom (n -1) when we compare both groups with ID from the validation sample (with ASD and without ASD). In any case, we have eliminated it as after reading the suggestions from the other editor we changed the analysis.

I would recommend that more arguments regarding the obtained results be given in the article, considering the fact that more categories of subjects were researched. A longer bibliography could help in this direction. Also, the results should be analyzed and the conclusions should be compared more from the perspective of the results of other studies, especially because they used instruments applied to subjects from other countries and different contexts.

Answer: Introduction and conclusions have been rewritten in this way.

Improving the quality of life of people with disabilities is an important objective in social policies systems; that is why it is very important to make a correct assessment in order to offer support to this population category, especially young people and children. The results of the study should better point out the need for these studies which would also be useful from the perspective of educational integration and social inclusion, especially for children and young people. I recommend looking at other articles as well, such as: 

https://doi.org/10.3390/su13137056

https://doi.org/10.3390/su13052558

https://doi.org/10.3390/su14084630.

Answer: We have added the first article proposed.

Discussion has been rewritten taking into account these suggestions.

The presentation of a section with reference to the limits of the study, would demonstrate honesty and frankness in the investigation, but also implications for future research.

Answer: We have added a specific paragraph about limitations and future research lines.

The conclusions of the study should show the importance and necessity of correctly assessing the needs of people with disabilities, with appropriate tools, to improve their quality of life, integration and social inclusion.

Answer: We have added a sentence regarding this information.

  • We also sent the manuscript to the proof-reading service recommended for the editors of this journal -

Round 2

Reviewer 1 Report

Dear authors, my previous comments have been addressed and I would recommend this paper for publication.

English quality is improved but there still some minor oversights that should be addressed